# Glycyrrhizin Interacts with TLR4 and TLR9 to Resolve *P. aeruginosa* Keratitis

**DOI:** 10.3390/pathogens11111327

**Published:** 2022-11-11

**Authors:** Mallika Somayajulu, Sharon A. McClellan, Shukkur Muhammed Farooq, Ahalya Pitchaikannu, Shunbin Xu, Linda Hazlett

**Affiliations:** Department of Ophthalmology, Visual and Anatomical Sciences, Wayne State University School of Medicine, 540 E. Canfield, Detroit, MI 48201, USA

**Keywords:** keratitis, GLY, TLR4, TLR9

## Abstract

This study tests the mechanism(s) of glycyrrhizin (GLY) protection against *P. aeruginosa* keratitis. Female C57BL/6 (B6), TLR4 knockout (TLR4KO), myeloid specific TLR4KO (mTLR4KO), their wildtype (WT) littermates, and TLR9 knockout (TLR9KO) mice were infected with *P. aeruginosa* KEI 1025 and treated with GLY or PBS onto the cornea after infection. Clinical scores, photography with a slit lamp, RT-PCR and ELISA were used. GLY effects on macrophages (Mϕ) and polymorphonuclear neutrophils (PMN) isolated from WT and mTLR4KO and challenged with KEI 1025 were also tested. Comparing B6 and TLR4KO, GLY treatment reduced clinical scores and improved disease outcome after infection and decreased mRNA expression levels in cornea for TLR4, HMGB1, and RAGE in B6 mice. TLR9 mRNA expression was significantly reduced by GLY in both mouse strains after infection. GLY also significantly reduced HMGB1 (B6 only) and TLR9 protein (both B6 and TLR4KO). In TLR9KO mice, GLY did not significantly reduce clinical scores and only slightly improved disease outcome after infection. In these mice, GLY significantly reduced TLR4, but not HMGB1 or RAGE mRNA expression levels after infection. In contrast, in the mTLR4KO and their WT littermates, GLY significantly reduced corneal disease, TLR4, TLR9, HMGB1, and RAGE corneal mRNA expression after infection. GLY also significantly reduced TLR9 and HMGB1 corneal protein levels in both WT and mTLR4KO mice. In vitro, GLY significantly lowered mRNA expression levels for TLR9 in both Mϕ and PMN isolated from mTLR4KO or WT mice after incubation with KEI 1025. In conclusion, we provide evidence to show that GLY mediates its effects by blocking TLR4 and TLR9 signaling pathways and both are required to protect against disease.

## 1. Introduction

*Pseudomonas aeruginosa* is an opportunist pathogen associated with contact lens wear and microbial keratitis worldwide [1,2,3]. We established a strong correlation between high-mobility group box 1 (HMGB1) and the severity of *P. aeruginosa* keratitis [4]. HMGB1 belongs to a family of danger-associated molecular patterns (DAMPS), which amplifies inflammation. It can be released actively or passively from cells [5]. HMGB1 can signal via binding to many receptors including the receptor for advanced glycation end products (RAGE) and Toll-like receptors (TLRs) such as TLR2, TLR4 and TLR9 to activate various inflammatory mediators [6]. These are important in the pathology of a variety of diseases, including ocular pathologies [4,7,8]. Intensive antibiotic therapy is used for treatment, but with emerging antibiotic resistance of *P. aeruginosa* and other pathogens, developing alternative or adjunctive therapeutics is needed [9,10,11]. To this end, it has been reported that GLY directly binds HMGB1 and abrogates its chemokine and cytokine mediated inflammatory response [11,12,13]. GLY, structurally a saponin, has anti-inflammatory properties and has been effectively used clinically to treat patients with viral hepatitis C [14]. Experimental studies from our laboratory have shown that GLY is effective against *P. aeruginosa* keratitis induced by a cytotoxic strain (ATCC 19660) [11], and an ocular noncytotoxic clinical isolate, KEI 1025 [15] particularly when combined with antibiotics. The exact mechanism by which GLY is effective is currently not completely understood. Thus, the objectives of the study are to test whether GLY inhibits HMGB1 amplification of TLR4 signaling pathways and if TLR9 activation has beneficial effects in *P. aeruginosa* keratitis.

## 2. Materials and Methods

### 2.1. Mice

Age matched (8 weeks old) female B6 and B6.B10ScN-*Tlr4^lps-del^*/JthJ (TLR4 knockout (KO), strain #007227) and C57BL/6-*Tlr9*^*em*1.1*Ldm*^/J (TLR9KO, stock #034449) mice were purchased from the Jackson Laboratory (Bar Harbor, ME) and housed in accordance with the National Institutes of Health guidelines. Myeloid specific TLR4KO (mTLR4KO) mice were also generated. To do this, three rounds of breeding were carried out. In the first round, male B6.129P2-Lyz2^tm1(Cre)Ifo^/J [LysM-Cre^(+/+)^] (Jackson Labs, stock #004781) were bred with female B6(Cg)-Tlr4^tm1.1Karp^/J [TLR4^f/f^] (Jackson Labs, stock # 024872) to produce LysM-Cre^(+/−)^;TLR4^f/+^ mice. In the second round of breeding, LysM-Cre^(+/−)^;TLR4^f/+^ mice were intercrossed to produce LysM-Cre^(−/−)^;TLR4^f/f^ and LysM-Cre^(+/−)^;TLR4^f/f^ that were bred in the third round to generate LysM-Cre^(+/−)^;TLR4^f/f^ [myeloid specific TLR4KO (mTLR4KO)] and LysM-Cre^(−/−)^;TLR4^f/f^ mice (WT, age matched littermates). Mice were compassionately treated and complied with the ARVO Statement for the Use of Animals in Ophthalmic and Vision Research and the Institutional Animal Care and Use Committee of Wayne State University (IACUC-21-04-3499).

### 2.2. Bacterial Culture and Infection

Bacteria were cultured using a previously published protocol [15,16]. Briefly, *P. aeruginosa* strain, KEI 1025, an ocular noncytotoxic clinical isolate (Kresge Eye Institute, Detroit, MI, USA), was grown in peptone tryptic soy broth (PTSB) medium in a rotary shaker water bath at 37 °C and 150 rpm for 18h to an optical density (measured at 540 nm) between 1.3 and 1.8. Bacterial cultures were centrifuged at 5500 g for 10min; pellets were washed once with sterile saline, re-centrifuged, re-suspended, and diluted in sterile saline. Mice were infected as previously described [16]. Briefly, mice were anesthetized (using anhydrous ethyl ether) and placed under a stereoscopic microscope at 40× magnification. The left cornea was scarified using a 25^5/8^gauge needle. The wounded corneal surface then received 5 μL containing 1 × 10^7^ colony-forming units (CFU)/μL of the *P. aeruginosa* KEI 1025 suspension applied topically. B6, TLR4KO, TLR9KO and the mTLR4KO mice were infected with KEI 1025 as described above.

### 2.3. Response to Bacterial Infection

A scale to grade disease was used to assign clinical scores to each eye at 1, 3, and 5 days post infection (p.i.) [17] and photographed 5 days p.i. with a slit lamp. Clinical scores were designated as follows: 0 = clear or slight opacity, partially or fully covering the pupil; +1 = slight opacity, fully covering the anterior segment; +2 = dense opacity, partially or fully covering the pupil; +3 = dense opacity, covering the entire anterior segment; and +4 = corneal perforation or phthisis. Each mouse was scored in masked fashion.

### 2.4. GLY Treatment after KEI 1025 Infection

The left eyes of B6, TLR4KO, TLR9KO and mTLR4KO mice infected with KEI 1025 were topically treated with 5 µL GLY (100 μg in 5 µL PBS) or PBS 6h after infection and twice daily from 1 to 4 days p.i. [15].

### 2.5. Real Rime RT-PCR

Mice that were infected with KEI 1025 and treated with GLY or PBS were killed at 5 days p.i. and the infected corneas were harvested. Total RNA from each cornea (n = 5/group/time) was isolated (RNA STAT-60; Tel-Test, Friendswood, TX, USA) as reported before [18]. One μg of each RNA sample was reverse transcribed using Moloney-murine leukemia virus (M-MLV) reverse transcriptase (Invitrogen, Carlsbad, CA, USA) to produce a cDNA template. cDNA products were diluted 1:20 with DEPC-treated water and a 2 μL aliquot of diluted cDNA was used for the RT-PCR reaction. SYBR green/fluorescein PCR master mix (Bio-Rad Laboratories, Richmond, CA, USA) and primer concentrations of 10 μM were used in a total 10 μL volume. After a preprogrammed hot start cycle (3 min at 95 °C), the parameters used for PCR amplification were: 15 s at 95 °C and 60 s at 60 °C with the cycles repeated 45 times. Levels of HMGB1, TLR4, TLR9 and RAGE were tested by real-time RT-PCR (CFX Connect real-time PCR detection system; Bio-Rad Laboratories, Hercules, CA, USA). The fold differences in gene expression were calculated relative to control and normalized to the housekeeping gene 18S ribosomal RNA (mouse) and expressed as the relative mRNA concentration + SEM. Primer pair sequences used are shown in Table 1.

### 2.6. ELISA

Mice that were KEI 1025 infected, and treated with GLY or PBS (n = 5/group/time) were euthanized at 5 days p.i. and infected corneas harvested in 500µL of PBS containing 0.1% Tween 20 and protease inhibitors. ELISA kits were used to detect HMGB1 (Chondrex, Inc., Redmond, WA, USA), and TLR9 (Novus Biologicals, Centennial, CO, USA) per the manufacturers’ protocol.

### 2.7. Mϕ Isolation 

A published protocol was used to isolate Mϕ from B6 and mTLR4KO mice [19,20]. Briefly, Mϕ were induced into the peritoneal cavity by intraperitoneal injection of 1 mL of 3% Brewer’s thioglycollate medium (BD Biosciences, Sparks, MD, USA) 5 days before euthanizing. Cells were collected by peritoneal lavage with DMEM containing 5% FBS. Viable cells (>95%) were counted with a hemocytometer by staining with 0.4% trypan blue. Mϕ were seeded into 6-well tissue culture plates at a density of 2.5 × 10^5^ cells/well in the absence and presence of 2 mM GLY, and were incubated for 3 h. Then, non-adherent cells were removed and cells were incubated with fresh media in the presence or absence of 2 mM GLY. Mϕ were incubated with KEI 1025 (Multiplicity of infection (MOI) 10) for 3 h at 37 °C. A subset of Mϕ were mock-treated with growth medium and incubated for 3 h at 37 °C to serve as the negative controls. RT-PCR determined mRNA levels of TLR4 and TLR9.

### 2.8. PMN Isolation

PMN were collected from bone marrow of B6 and mTLR4KO mice as previously described [21]. Briefly, the femurs and tibia were removed from the mice after euthanasia and flushed with Hanks buffered Salt Solution (HBSS, ThermoFisher Scientific, Waltham, MA, USA). Cell clumps were dispersed, debris removed and centrifuged. Pellets were washed, resuspended in HBSS, and layered on top of a three-step discontinuous Percoll (Cytiva Sweden AB, Uppsala, SWE, USA) density gradient prepared in a 15 mL polystyrene tube by layering 2 mL of 75, 67, and 52% Percoll solutions. PMN fraction was collected from the lowest band (at the 75/67% interface) after centrifugation at 1647 *g* for 30 min at room temperature. Red blood cells were eliminated by hypotonic lysis. The purity of PMN was typically above 90%, assessed by Leishman Stain (Sigma, St. Louis, MO, USA). PMN were treated with 2 mM GLY and incubated with KEI 1025 (MOI 10) for 1.5 h as described above and mRNA levels of TLR4 and TLR9 were determined by RT-PCR.

### 2.9. Statistical Analysis

A 1-way ANOVA followed by the Bonferroni’s multiple comparison test (GraphPad Prism, San Diego, CA, USA) was used for analysis when comparing three or more groups (RT-PCR and ELISA). Clinical scores, compared between two groups at each time, were tested by the Mann–Whitney U test. For each test, *p* < 0.05 was considered significant and data are shown as mean + SEM. All experiments were repeated at least once to ensure reproducibility.

## 3. Results

### 3.1. Effects of GLY on B6 and TLR4KO Corneas

Data shown in Figure 1A exhibit clinical scores of KEI 1025 infected B6 and TLR4KO mice after treatment with GLY vs. PBS. No significant change in clinical scores was seen after application of GLY vs. PBS in either B6 or TLR4KO mice at 1 day p.i. Significantly less disease with lower clinical scores was seen only in the corneas of B6 mice after application of GLY vs. PBS at 3 (*p* < 0.05) and 5 (*p* < 0.01) days p.i., respectively. Photographs taken with a slit lamp of representative corneas from PBS treated B6 mice (Figure 1B). GLY treated B6 mice (Figure 1C), PBS treated TLR4KO mice (Figure 1D) and GLY treated TLR4KO mice (Figure 1E) at 5 days p.i. Data show that the majority of the eyes treated with PBS (both B6 and TLR4KO) were perforated. No perforation was observed in the GLY treated B6 eyes that also exhibited reduced opacity confined to the central cornea with hypopyon present in the anterior chamber. GLY vs. PBS treated TLR4KO mice showed corneal thinning, but no perforation, and closely compacted opacity covering the anterior segment.

### 3.2. GLY Effects on mRNA and Protein Levels in B6 and TLR4KO Corneas

The relative mRNA expression levels for TLR4 (A), TLR9 (B), HMGB1 (C) and RAGE (D) 5 days after B6 and TLR4KO mice were infected with KEI 1025 and treated with GLY or PBS are shown in Figure 2. GLY vs. PBS significantly reduced TLR4 (A, *p* < 0.001), HMGB1 (C, *p* < 0.001), and RAGE (D, *p* < 0.001) mRNA expression levels only in infected B6 mice. mRNA expression of HMGB1 was only slightly reduced and RAGE slightly increased in infected TLR4KO mice by GLY treatment. TLR4 mRNA was undetectable in the TLR4KO mice. In both infected B6 and TLR4KO mice, GLY vs. PBS significantly reduced TLR9 (B, *p* < 0.001, *p* < 0.05) mRNA expression levels. 

Protein analysis (Figure 3) showed GLY significantly lowered TLR9 (A, *p* < 0.05, *p* < 0.01) protein levels in both B6 and TLR4KO 5 days after infection, respectively. However, significant differences in HMGB1 protein levels between PBS and GLY treated groups were only seen at 5 days p.i. in infected B6 but not infected TLR4KO mice (B, *p* < 0.001).

### 3.3. Effects of GLY on TLR9KO Corneas Infected with KEI 1025

KEI 1025 infected B6 and TLR9KO mice after treatment with GLY vs. PBS are shown in Figure 4. No significant change in clinical scores was seen after application of GLY vs. PBS in either B6 or TLR9KO at 1 day p.i. Significantly less corneal disease with lower clinical scores was seen only in B6 but not TLR9KO mice after application of GLY vs. PBS at both 3 (*p* < 0.01) and 5 (*p* < 0.01) days after infection. Photographs obtained from slit lamp photography of corneas from PBS treated B6 mice (Figure 4B), GLY treated B6 mice (Figure 4C), PBS treated TLR9KO mice (Figure 4D) and GLY treated TLR9KO mice (Figure 4E) are shown at 5 days p.i. Data provide evidence that the majority of eyes treated with PBS (both B6 and TLR9KO) were perforated. GLY treated B6 mouse eyes showed no perforation and exhibited reduced opacity confined to the central cornea with hypopyon present in the anterior chamber. GLY treated TLR9KO mice displayed corneal thinning but no perforation and dense opacity covered the anterior segment compared to the PBS treated TLR9KO. 

### 3.4. GLY Effects on TLR, HMGB1 and RAGE Levels in B6 and TLR9KO Corneas 

The relative mRNA levels for TLR4 (A), TLR9 (B), HMGB1 (C) and RAGE (D) 5 days after B6 and TLR9KO mice were infected with KEI 1025 and treated with GLY or PBS are shown in Figure 5. GLY vs. PBS significantly reduced TLR4 (A, *p* < 0.001, *p* < 0.001) mRNA expression levels in both infected B6 mice and TLR9KO mice. However, GLY vs. PBS significantly reduced HMGB1 (C, *p* < 0.001), and RAGE (D, *p* < 0.001) mRNA expression only in infected B6 mice. mRNA levels of HMGB1 and RAGE were slightly, but not significantly reduced in infected TLR9KO mice. As expected, GLY reduced TLR9 mRNA expression in B6 mice (B, *p* < 0.001) and TLR9 was undetectable in the TLR9KO mice. 

### 3.5. Effects of GLY on mTLR4KO Corneas 

TLR4 levels were assessed in Mϕ and PMN from mTLR4KO and were, as anticipated, undetectable (Figure 6A,B). Figure 6C shows clinical scores of KEI 1025 infected WT littermates and mTLR4KO mice after treatment with GLY vs. PBS. Significantly reduced clinical scores were observed in GLY vs. PBS treated WT littermates and mTLR4KO mice infected with KEI 1025 at 3 (*p* < 0.01, *p* < 0.01) and 5 (*p* < 0.001, *p* < 0.01) days p.i. Photographs taken with a slit lamp of representative corneas from PBS treated WT littermate mice (Figure 6D), GLY treated WT littermate mice (Figure 6E), PBS treated mTLR4KO mice (Figure 6F) and GLY treated mTLR4KO mice (Figure 6G) at 5 days p.i. Data show that the majority of the WT littermate mouse eyes treated with PBS (D) exhibited corneal perforation or thinning. In GLY treated WT littermate eyes, no perforation and only light opacity covering the anterior segment was observed (E). In PBS treated mTLR4KO eyes (F), perforation was observed along with dense opacity covering the entire cornea. GLY treated mTLR4KO eyes (G) showed opacity confined to the central cornea and hypopyon in the anterior segment.

### 3.6. GLY Effects on mRNA and Protein Levels in WT Littermates and mTLR4KO Corneas

Relative mRNA levels for TLR4 (A), TLR9 (B), HMGB1 (C) and RAGE (D) at 5 days after WT littermates and mTLR4KO mice were infected with KEI 1025 and treated with GLY or PBS are shown in Figure 7. In both infected WT littermate and mTLR4KO mice, GLY vs. PBS significantly reduced corneal mRNA levels for TLR4 (A, *p* < 0.01, *p* < 0.01), TLR9 (B, *p* < 0.01, *p* < 0.05), HMGB1 (C, *p* < 0.05, *p* < 0.01), and RAGE (D, *p* < 0.001, *p* < 0.001). 

Protein studies (Figure 8) showed significant differences in TLR9 (A, *p* < 0.001, *p* < 0.05) and HMGB1 (B, *p* < 0.001, *p* < 0.001) levels between PBS and GLY treated groups 5 days p.i. in both WT littermates and mTLR4KO mice.

### 3.7. Effects of GLY on TLR Levels in Mϕ and PMN Isolated from WT Littermate and mTLR4KO Mice Incubated with KEI 1025 In Vitro

The relative mRNA levels for TLR4 (A,C) and TLR9 (B,D) in Mϕ (A,B) and PMN (C,D) isolated from WT littermate and mTLR4KO mice, incubated with KEI 1025 with or without GLY is shown in Figure 9. mRNA levels of TLR4 (A, *p* < 0.001) were significantly reduced by GLY in only WT littermate Mϕ incubated with KEI 1025, while remaining undetectable in mTLR4KO Mϕ. However, TLR9 was significantly reduced by GLY in both WT littermate and mTLR4KO Mϕ (B, *p* < 0.001, *p* < 0.001) incubated with KEI 1025. PMN isolated from WT littermate and mTLR4KO mice, incubated with KEI 1025 and treated with or without GLY showed TLR4 mRNA levels were significantly reduced (C, *p* < 0.001) by GLY in WT littermate cells and remained undetectable in PMN isolated from mTLR4KO. However, GLY significantly reduced levels of TLR9 (D, *p* < 0.001, *p* < 0.001) in PMN derived from both WT littermate and mTLR4KO mice incubated with KEI 1025.

## 4. Discussion

The innate immune system plays an important role in the initial detection of microbes by employing pattern recognition receptors (PRRs) [22]. Among these PRRs, TLRs are the best characterized [22]. TLR4, recognizes Gram-negative lipopolysaccharide (LPS) as well as endogenous molecules released during tissue damage [18,23]. In this regard, HMGB1, an alarmin and a member of the family of DAMPS, also binds to TLR4 with high affinity [24], and to RAGE [25] and TLR9 [26] to induce expression of several cytokines and chemokines [25,27,28]. In previous studies, GLY was identified as a potent inhibitor of HMGB1 [29]. It directly binds to it by interacting with two of the three functional domains of HMGB1 [30], to inhibit the chemotactic function of HMGB1 in a variety of diseases [11,30,31,32]. In *P. aeruginosa* induced keratitis, for example, GLY binds to HMGB1, inhibits inflammation and decreases bacterial load [11] to provide better disease outcome. However, it is unclear if it is through TLR4 alone or whether TLR9 is involved. In the present study, we tested whether GLY protects the cornea against *P. aeruginosa* induced keratitis by blocking the HMGB1-TLR4 pathway exclusively or if TLR9 is involved. 

TLR4 activation is a double-edged sword, having different outcomes in different models [33,34]. For example, while TLR4 is crucial for the pathologic progression observed in noninfectious keratitis [35], it is protective in bacterial keratitis where it is often required for host resistance [18]. Previously, we have shown that TLR4 deficient BALB/c mice showed increased corneal opacity, and corneal perforation which manifests itself by increasing susceptibility to keratitis, as compared to WT control mice [18]. However, we had not explored infection in B6 mice that are susceptible (cornea perforates) to infection [4] despite the presence of TLR4. When testing the effects of GLY, we asked whether the presence or absence of TLR4 affected the outcome of GLY mediated protection. GLY treatment of B6 TLR4KO mice failed to exert protective effects in the corneas of these mice. GLY did not improve disease outcome, nor reduce corneal levels of pro-inflammatory molecules such as HMGB1 or RAGE, whereas in the presence of TLR4 it was effective. In contrast, when we examined TLR9 protein levels in infected B6 TLR4KO mouse corneas, we found that GLY was equally capable of reducing TLR9 as it did in B6 mice. This led us to postulate that in the absence of TLR4, GLY may exert its effects by modulating TLR9. 

To further understand the role of TLR9 in GLY mediated protection, we infected TLR9KO mice and treated them with GLY. In these corneas, GLY was unable to improve disease outcome statistically, although perforation was reduced (Figure 4). These data agree with previous studies where we provided evidence that TLR9 silencing in B6 mice (TLR4 was present in these mice) also reduced perforation, but did not reduce bacterial load in the cornea [36]. The requirement of TLR9 for effective innate immune responses has been reported in a TLR9 deficient mouse model on a BALB/c background that could not elicit an effective Th1 cytokine response after infection with Gram-negative bacterial pathogens such as *K. pneumoniae* [32]. Additionally, intratracheal administration of the bacteria in TLR9KO mice led to a reduced accumulation and maturation of dendritic cells, as well as impaired activation of macrophages in the lung [37]. We also observed TLR4 mRNA upregulation in TLR9KO mouse corneas (Figure 5). These data are similar to previous studies that have shown a reciprocal expression (and signaling) through TLR9 and TLR4 in necrotizing enterocolitis [38]. In regard to GLY mediated protection, in the current study, GLY was unable to reduce the levels of RAGE and HMGB1 in the absence of TLR9, but significantly reduced TLR4 levels in TLR9KO corneas after infection. These data suggest that TLR4 is the key regulator in GLY mediated protection but that for full effect (reduction of HMGB1 and RAGE); it requires the presence of TLR9. To provide further evidence that GLY mediates its effects primarily through TLR4, but also requires TLR9 as a secondary modulator, we selectively depleted TLR4 from myeloid cells using a Cre-LoxP system [39]. In these mice, GLY was able to protect the cornea by lowering clinical scores at both 3 and 5 days after infection (Figure 6) and downregulating TLR9, HMGB1 and RAGE after infection. We postulate that GLY binds to extracellular HMGB1 and thus renders it inaccessible for binding to its receptors. In this regard, GLY can bind to HMGB1 and thus block its interaction with TLR4 and RAGE [31]. These data provide confirmation that TLR4 together with TLR9 are required for GLY to exert its protective effects. Additionally, in previous studies, we have shown that following infection, there is a predominant PMN infiltrate into the corneal stroma, but a small percentage of the infiltrate is the Mϕ [40]. Mϕ are important cells which we have shown prevent bacterial growth and control immune responses by controlling PMN ingress and balancing pro- and anti-inflammatory cytokines [27,41]. When LPS or Gram-negative bacteria bind to TLR4 on Mϕ, they activate the production of pro-inflammatory cytokines such as IL-6, IL-1β and TNF-α [42]. Therefore, we examined whether GLY had an effect on PMN and Mϕ in which TLR4 was specifically depleted (mTLR4KO). PMN and Mϕ, incubated with KEI 1025, showed that GLY reduced TLR9 levels in both cell types, suggesting that GLY mediates its protective effects primarily by blocking HMGB1-TLR4 signaling pathways, but for full effect, it requires the presence of TLR9. Thus, in conclusion we have provided evidence to show that GLY mediates its effects by blocking TLR4 and TLR9 signaling pathways and both are required to protect against disease (Figure 10).

## Figures and Tables

**Figure 1 pathogens-11-01327-f001:**
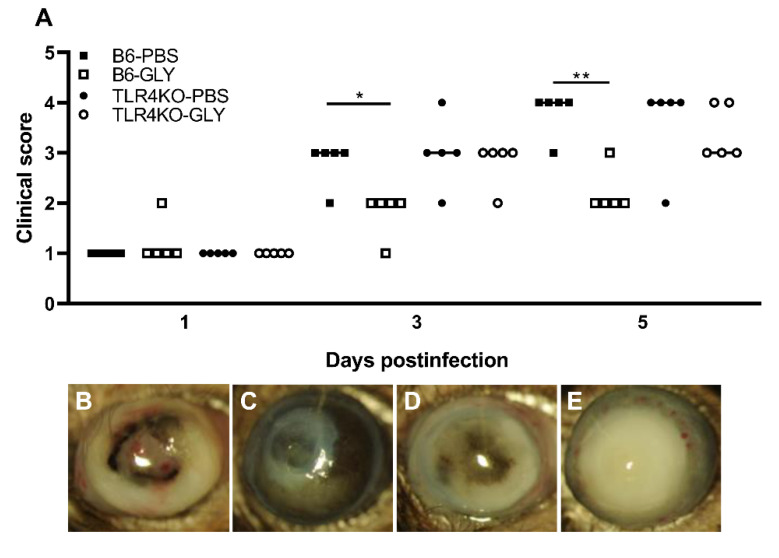
Treatment of B6 and TLR4KO mice with GLY after infection with KEI 1025. Topical treatment with 20 mg/mL GLY (5 μL, twice daily) significantly reduced disease scores at 3 and 5 days p.i. in B6, but not TLR4KO mice when compared to PBS treatment (**A**). Slit lamp photographs (**B**–**E**) show perforation in corneas of both B6 and TLR4KO mice at 5 days p.i. ((**B**,**D**), respectively). GLY treated B6 mouse corneas showed opacity over the pupil and hypopyon (**C**), compared to dense opacity covering the entire anterior chamber in the TLR4KO mice (**E**). Data analyzed using a Mann–Whitney U test. (n = 5/group/time). * *p* < 0.05, ** *p* < 0.01.

**Figure 2 pathogens-11-01327-f002:**
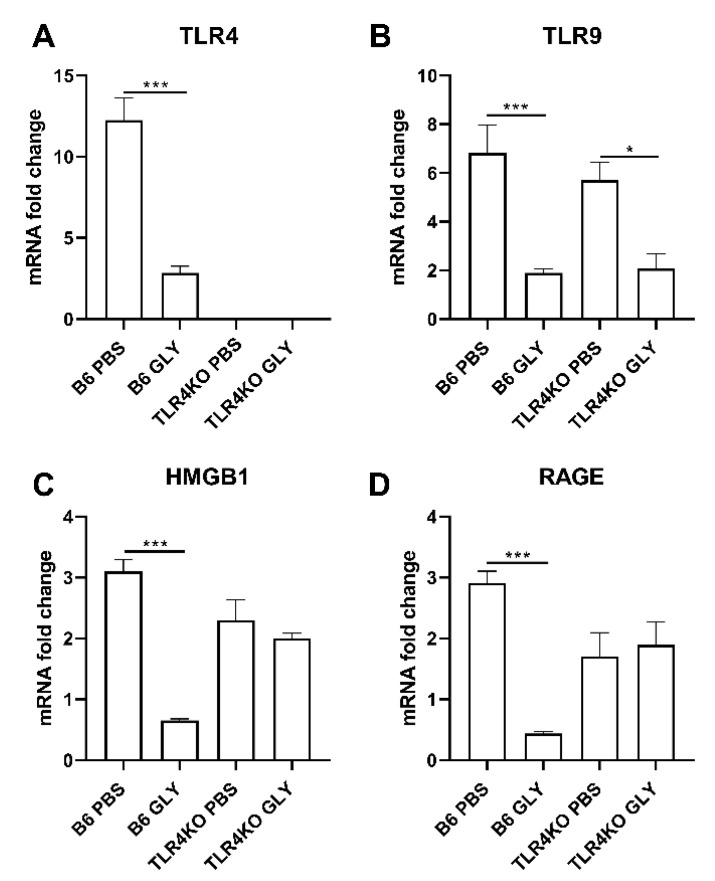
mRNA expression levels of TLRs, HMGB1 and RAGE (**A**–**D**). After treatment of KEI 1025 infected corneas with PBS or GLY, mRNA expression levels for TLR4 (**A**), TLR9 (**B**), HMGB1 (**C**), and RAGE (**D**) were significantly reduced in the cornea of GLY treated B6 mice compared to PBS treatment. Only TLR9 mRNA expression was reduced by GLY treatment in TLR4KO mice. Data are mean + SEM analyzed using 1-way ANOVA followed by the Bonferroni’s multiple comparison test. (n = 5/group/time). * *p* < 0.05, *** *p* < 0.001.

**Figure 3 pathogens-11-01327-f003:**
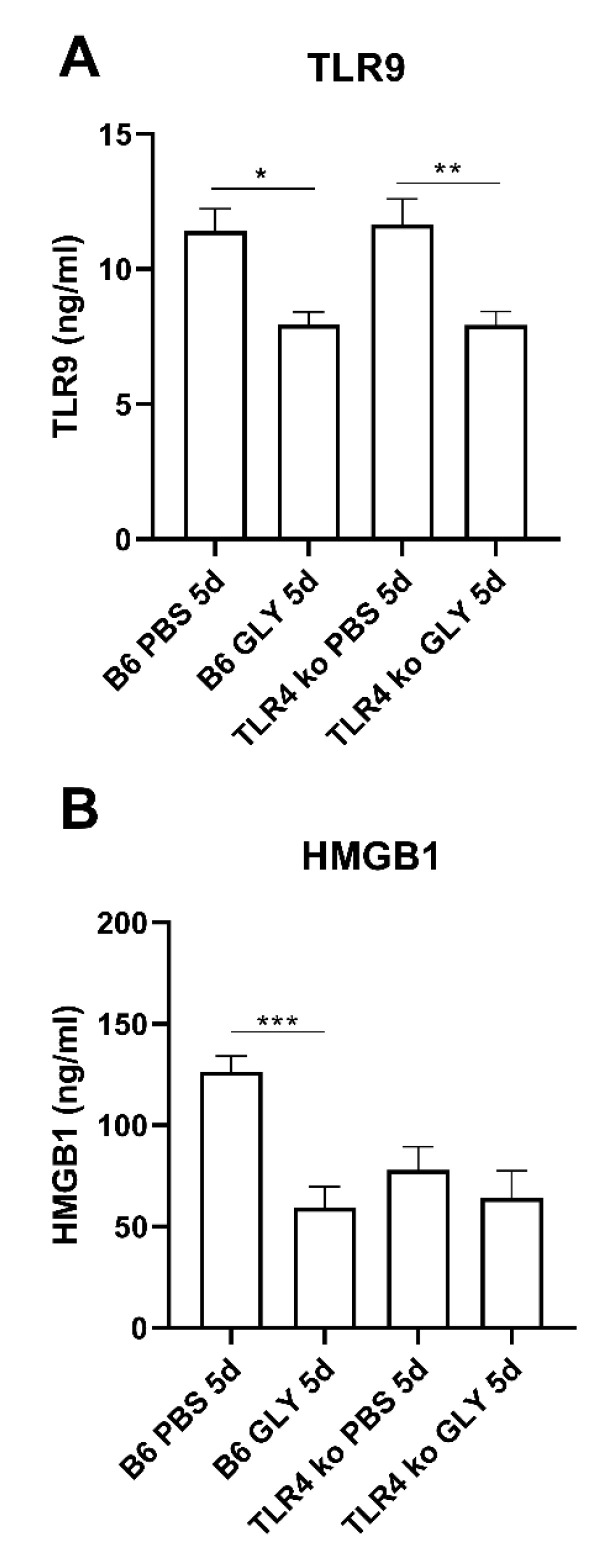
ELISA assay for TLR9 and HMGB1 protein. ELISA assays confirmed mRNA expression data and showed GLY treatment significantly reduced TLR9 (**A**) and HMGB1 (**B**) protein levels in B6 corneas compared to PBS treatment, but only TLR9 protein levels were reduced by GLY in the TLR4KO mouse cornea. Data are mean + SEM analyzed using 1-way ANOVA followed by the Bonferroni’s multiple comparison test. (n = 5/group/time). * *p* < 0.05, ** *p* < 0.01, *** *p* < 0.001.

**Figure 4 pathogens-11-01327-f004:**
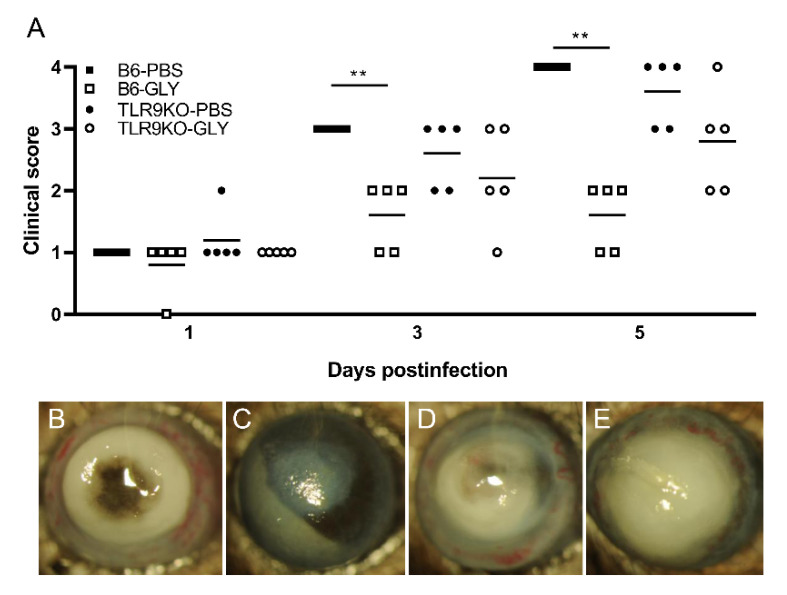
Treatment of B6 and TLR9KO mice with GLY after infection with KEI 1025. TLR9KO and B6 mice were infected with *P. aeruginosa*, KEI 1025. Topical treatment with 20 mg/mL GLY (5 μL, twice daily) vs. PBS significantly decreased disease scores at 3 and 5 days p.i. in B6 mice, but not TLR9KO (**A**). Slit lamp photographs display perforation in corneas of both B6 and TLR9KO at 5 days p.i. ((**B**,**D**), respectively). GLY treated B6 corneas showed opacity over the pupil and hypopyon (**C**), compared to dense opacity covering the entire anterior chamber in the TLR9KO (**E**). Data analyzed using a Mann–Whitney U test. (n = 5/group/time). ** *p* < 0.01.

**Figure 5 pathogens-11-01327-f005:**
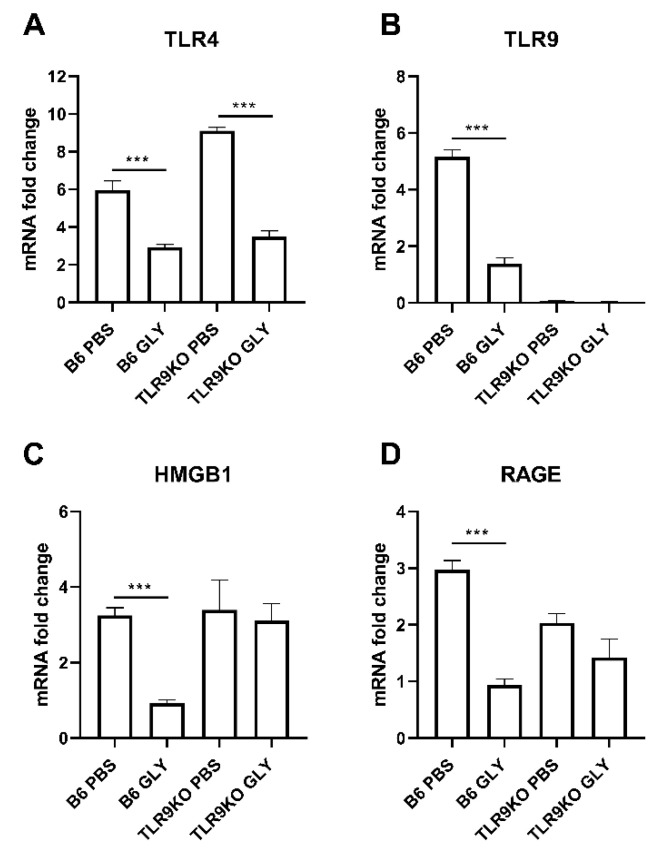
mRNA expression of TLRs, HMGB1 and RAGE (**A**–**D**). After treatment of KEI 1025 infected corneas with PBS or GLY, mRNA expression levels for TLR4 (**A**), TLR9 (**B**), HMGB1 (**C**)**,** and RAGE (**D**) were significantly reduced in the cornea of GLY vs. PBS treated B6 mice. Only TLR4 mRNA expression was reduced by GLY treatment in TLR9KO. Data are mean + SEM analyzed using 1-way ANOVA followed by the Bonferroni’s multiple comparison test. (n = 5/group/time). *** *p* < 0.001.

**Figure 6 pathogens-11-01327-f006:**
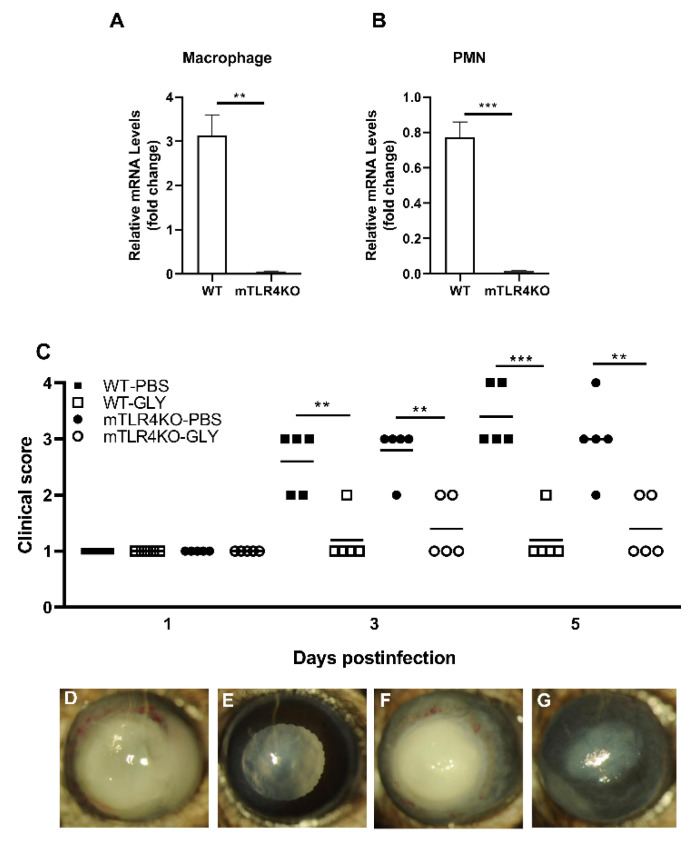
Confirmation of TLR4KO and treatment of mTLR4KO mice and their WT littermates with GLY after infection with KEI 1025. RT-PCR was used to confirm the absence of TLR4 in both the macrophage (**A**) and PMN (**B**) cells from mTLR4KO mice. mTLR4KO and wild type littermate mice were infected with *P. aeruginosa*, KEI 1025. Topical treatment with 20 mg/mL GLY (5 μL, twice daily) significantly reduced disease scores at 3 and 5 days p.i. in both mTLR4KO and wild type littermate mice (**C**). Slit lamp photographs show perforation in corneas of wild type littermates and mTLR4KO at 5 days p.i. ((**D**,**F**), respectively). GLY treated wild type littermate corneas showed slight opacity over the pupil (**E**), compared to a denser opacity over the pupil with hypopyon in the anterior chamber in GLY treated mTLR4KO mice (**G**). RT-PCR data was analyzed using an unpaired Student’s *t*-test and clinical score data analyzed using a Mann–Whitney U test. (n = 5/group/time). ** *p* < 0.01, *** *p* < 0.001.

**Figure 7 pathogens-11-01327-f007:**
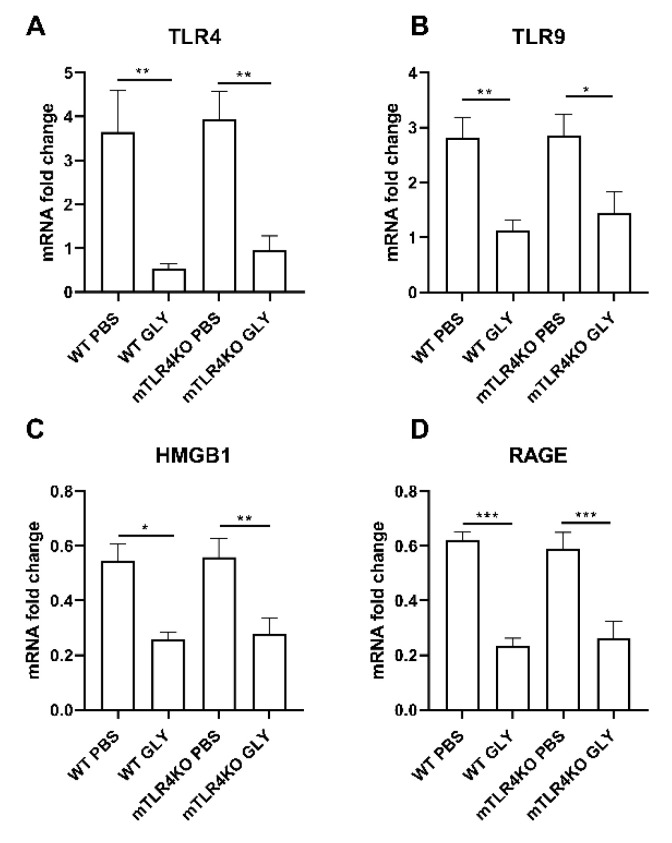
mRNA expression of TLRs, HMGB1 and RAGE (**A**–**D**). RT-PCR analysis revealed that GLY treatment reduced mRNA expression of TLR4, TLR9, HMGB1, and RAGE in WT and mTLR4KO mice compared to PBS treatment. Data are mean + SEM analyzed using 1-way ANOVA followed by the Bonferroni’s multiple comparison test. (n = 5/group/time). * *p* < 0.05, ** *p* < 0.01, *** *p* < 0.001.

**Figure 8 pathogens-11-01327-f008:**
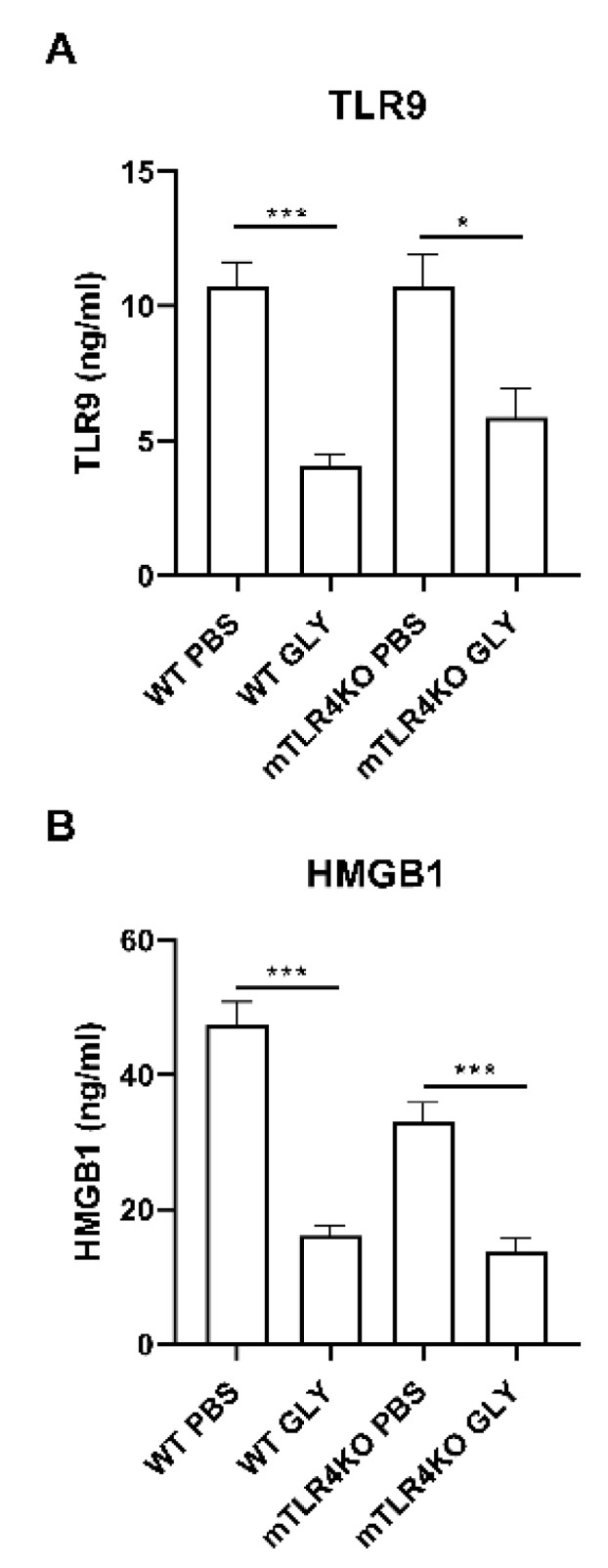
ELISA assay for TLR9 and HMGB1 protein. ELISA assays confirmed mRNA expression data and showed GLY treatment significantly reduced TLR9 (**A**) and HMGB1 (**B**) protein levels in WT littermates and mTLR4KO corneas compared to PBS treatment. Data are mean + SEM analyzed using 1-way ANOVA followed by the Bonferroni’s multiple comparison test. (n = 5/group/time). * *p* < 0.05, *** *p* < 0.001.

**Figure 9 pathogens-11-01327-f009:**
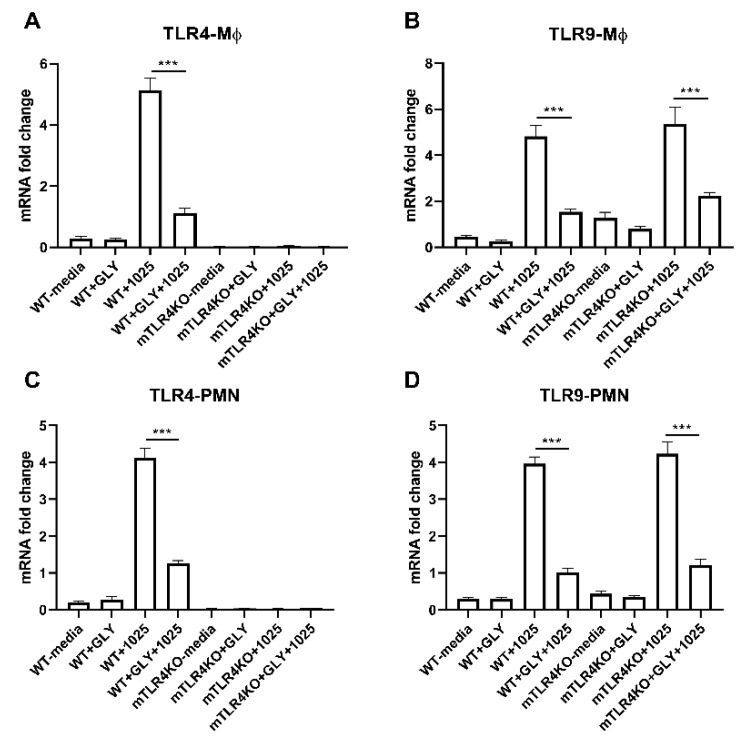
mRNA expression of TLR4 and TLR9 in Mϕ and PMN. mRNA expression levels for TLR4 (**A**,**C**) and TLR9 (**B**,**D**) in Mϕ (**A**,**B**) and PMN (**C**,**D**) isolated from wild type littermates and mTLR4KO mice, incubated with KEI 1025 in the presence or absence of GLY. mRNA levels of TLR4 (**A,C**) were significantly reduced by GLY only in cells from wild type littermate mice incubated with KEI 1025, while remained undetectable in cells from mTLR4KO. However, TLR9 was significantly reduced by GLY in both Mϕ (**B**) and PMN (**D**) from wild type littermates and mTLR4KO (**B**,**D**) mice incubated with KEI 1025. Data are mean + SEM analyzed using 1-way ANOVA followed by the Bonferroni’s multiple comparison test. (n = 5/group/time). *** *p* < 0.001.

**Figure 10 pathogens-11-01327-f010:**
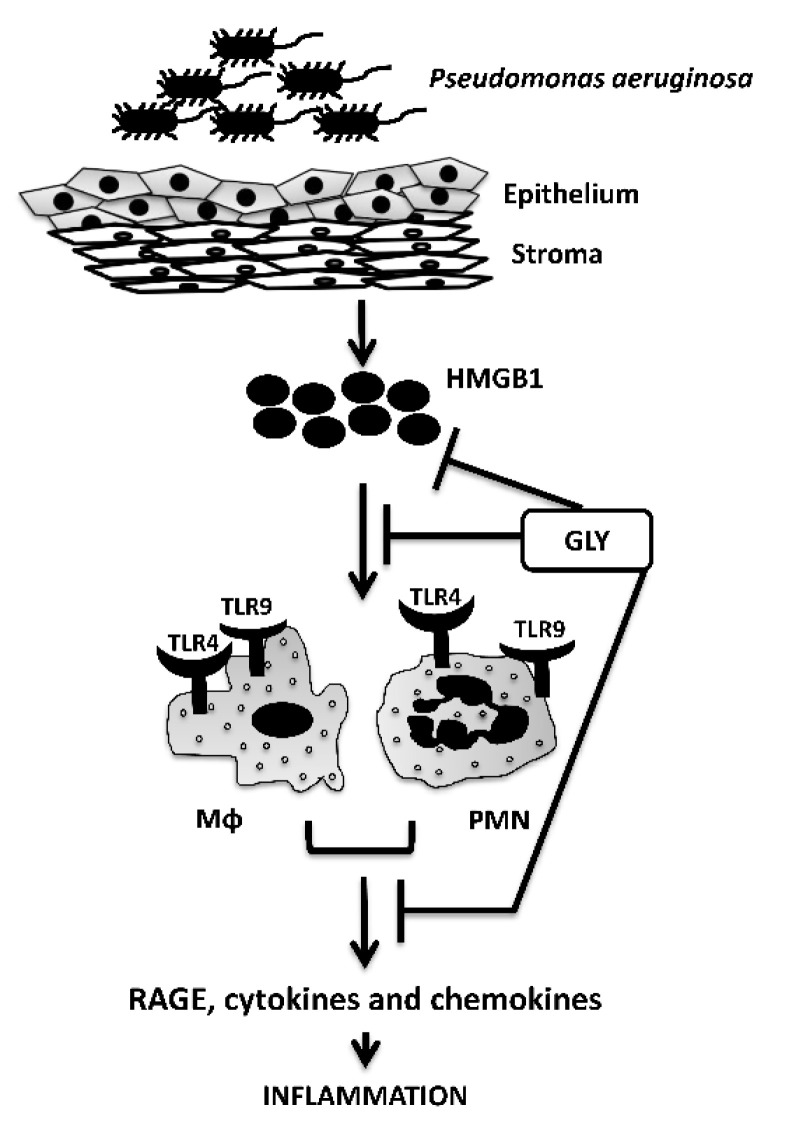
Summary diagram of response to infection.

**Table 1 pathogens-11-01327-t001:** Nucleotide sequence of the specific primers used for PCR amplification.

Gene	Nucleotide Sequence	Primer	GenBank
*18s*	5′-GTA ACC CGT TGA ACC CCA TT-3′ 5′-CCA TCC AAT CGG TAG TAG CG-3′	F R	NR_003278.3
*Tlr9*	5′-AGC TCA ACC TGT CCT TCA ATT ACC GC-3′ 5′-ATG CCG TTC ATG TTC AGC TCC TGC-3′	F R	NM_031178.2
*Tlr4*	5′-CGC TTT CAC CTC TGC CTT CAC TAC AG-3′ 5′-ACA CTA CCA CAA TAA CCT TCC GGC TC-3′	F R	NM_021297.2
*Rage*	5′-AGG CGT GAG GAG AGG AAG GCC-3′	F R	NM_007425.2
	5′-TTA CGG TCC CCC GGC ACC AT-3′		
*Hmgb1*	5′-TGG CAA AGG CTG ACA AGG CTC-3′	F R	NM_010439.3
	5′-GGA TGC TCG CCT TTG ATT TTG G-3′		
F, forward, R, reverse			

## Data Availability

Not applicable.

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
