# Peer review of "Glycyrrhizin Interacts with TLR4 and TLR9 to Resolve P. aeruginosa Keratitis"

_pathogens, 2022, doi:10.3390/pathogens11111327_

Round 1

Reviewer 1 Report

The ms by Somayajulu et al studied the effect of glycyrrhizin on Pseudomonas keratitis in continuation of earlier works. Here are some comments,

1. The title needs to be more descriptive.

2. What is the rationale for using the clinical isolate when earlier work was done with the ATCC strain?

3. Although the effect of Gly on TLR4 and 9 was tested, the authors did not look into the downstream signalling pathways. It would be beneficial to know the effect of Gly on NfkB and MAPK pathways in these knockouts.

4. Although the heading of the 3.7 in results section says, "effects of Gly on cytokine level----", no data on cytokines has been showed and therefore it becomes very difficult to understand.

Reviewer 2 Report

The rationale for this study was to investigate the signalling pathways inhibited by glycorrhizin  in corneas infected with Pseudomonas aeruginosa (ocular keratitis)

The authors produce evidence that  glycorrhizin,   which is known to  bind to and block the activity of the alarmin HMGB1, mediates its effects by blocking both the TLR4 and TLR9 signalling pathways

The authors have produce convincing data that supports their conclusion based on clinical scores, mRNA and cytokine  expression levels

The paper is very well written, the text is succinct and clear, and the figures and their legends give easy access to the data. The text is devoid of typographical errors and is grammatically correct

The one issue that I had was the lack of a clear  rationale for using TLR4KO mice and the myeloid specific TLR4 KO (reduced expression?). To a microbiologist who may not be au fait with the cell biology and transgene technology the rationale for the  latter experiment    is not clear

The accessibility of the paper to a wider audience would be improved with a summary diagram

Could the authors comment on the clinical significance of glycorrhizin and whether there is any prospect of it being used to treat infections. Does the molecule have any effect of keratitis produced by other ocular pathogens eg Staphylococcus aureus

Round 2

Reviewer 1 Report

All comments are answered satisfactorily.